# Dysbiosis: An Indicator of COVID-19 Severity in Critically Ill Patients

**DOI:** 10.3390/ijms232415808

**Published:** 2022-12-13

**Authors:** Silvia Cuenca, Zaida Soler, Gerard Serrano-Gómez, Zixuan Xie, Jordi Barquinero, Joaquim Roca, Jose-Maria Sirvent, Chaysavanh Manichanh

**Affiliations:** 1Intensive Care Department (ICU), University Hospital of Girona Dr. Josep Trueta, Avda. França s/n, 17007 Girona, Spain; 2Medicine Department, Autonomous University of Barcelona (UAB), 08193 Cerdanyola del Vallès, Spain; 3Gut Microbiome Group, Vall d’Hebron Institut de Recerca (VHIR), Vall d’Hebron Hospital Universitari, Vall d’Hebron Barcelona Hospital Campus, Passeig Vall d’Hebron 119-129, 08035 Barcelona, Spain; 4Gene and Cell Therapy, VHIR, Vall d’Hebron Hospital Campus, Passeig Vall d’Hebron 119-129, 08035 Barcelona, Spain; 5Molecular Biology Institute of Barcelona (IBMB), Spanish National Research Council (CSIC), 08028 Barcelona, Spain; 6Institute of Biomedical Research of Girona Dr. Josep Trueta (IDIBGI), 17190 Salt, Spain

**Keywords:** gut and lung, bacterial and fungal microbiome, composition and load, severe COVID-19 cases in ICU, mechanical ventilation

## Abstract

Here, we examined the dynamics of the gut and respiratory microbiomes in severe COVID-19 patients in need of mechanical ventilation in the intensive care unit (ICU). We recruited 85 critically ill patients (53 with COVID-19 and 32 without COVID-19) and 17 healthy controls (HCs) and monitored them for up to 4 weeks. We analyzed the bacterial and fungal taxonomic profiles and loads of 232 gut and respiratory samples and we measured the blood levels of Interleukin 6, IgG, and IgM in COVID-19 patients. Upon ICU admission, the bacterial composition and load in the gut and respiratory samples were altered in critically ill patients compared with HCs. During their ICU stay, the patients experienced increased bacterial and fungal loads, drastic decreased bacterial richness, and progressive changes in bacterial and fungal taxonomic profiles. In the gut samples, six bacterial taxa could discriminate ICU-COV(+) from ICU-COV(−) cases upon ICU admission and the bacterial taxa were associated according to age, PaO2/FiO2, and CRP levels. In the respiratory samples of the ICU-COV(+) patients, bacterial signatures including *Pseudomonas* and *Streptococcus* were found to be correlated with the length of ICU stay. Our findings demonstrated that the gut and respiratory microbiome dysbiosis and bacterial signatures associated with critical illness emerged as biomarkers of COVID-19 severity and could be a potential predictor of ICU length of stay. We propose using a high-throughput sequencing approach as an alternative to traditional isolation techniques to monitor ICU patient infection.

## 1. Introduction

The World Health Organization (WHO) has reported more than 636 million cumulative cases of COVID-19 and 6.59 million deaths worldwide up to November 2022. After the first cases of COVID-19 were diagnosed in December 2019, there was increasing interest in the role of the microbiome in the context of patients admitted to the intensive care unit (ICU). While 40% of those with confirmed COVID-19 present no symptoms, others exhibit a wide range of clinical manifestations, ranging from mild symptoms to severe illness [1]. Published studies indicate that approximately 14% of hospitalized patients undergo a severe course of the disease requiring intensive care, and approximately 5 to 12% require mechanical ventilation in the ICU [2,3]. The reasons some patients develop respiratory failure while others have minimal symptoms or even no symptoms remain unclear [4]. SARS-CoV-2 infection may lead to acute respiratory distress syndrome (ARDS), which progresses from respiratory failure to death. As ACE2 (angiotensin-converting enzyme 2), identified as the functional receptor of SARS-CoV-2, is abundant in the epithelia of the respiratory and intestinal tracts, these two localizations are considered the two major sites of viral replication [5,6,7].

Dicker et al. associated *Pseudomonas*, Enterobacteriaceae, and *Stenotrophomonas* in sputum with severe bronchiectasis and greater lung inflammation [8]. The findings by Ren et al. [9] revealed that *Streptococcus* was enriched in recovered patients, while patients with a depleted *Streptococcus* population showed more dysbiotic microbiota. Yeoh et al. [10] demonstrated an association between the gut bacteria and poor or fatal clinical outcomes in hospitalized patients with COVID-19, and Zuo et al. [11] found an enrichment of fungal species assigned to *Candida* and *Aspergillus* in these patients.

Concerning the respiratory microbiome in COVID-19 patients, most studies have examined the upper respiratory tract using oropharyngeal [12] and nasopharyngeal swabs. However, few studies have addressed the microbiome of the lower airways by bronchoalveolar lavage (BAL) or using tracheal samples. Among these few studies, Sulaiman et al. [13] reported that *Mycoplasma salivarium* in the BAL of critically ill patients is associated with poor clinical outcomes.

The aims of this exploratory study were to (a) define baseline dysbiosis in ICU-COV(+) compared to ICU-COV(−), (b) evaluate the stability of the gut and tracheal aspirate microbiome (bacteria and fungi) over time, (c) and investigate the microbial signatures associated with biological and clinical features.

## 2. Results

### 2.1. Patients’ Characteristics

To study the association between the gut and respiratory microbiomes and critical illness in the context of COVID-19, three groups of participants were enrolled: critically ill patients with PCR-confirmed SARS-CoV-2 (referred to herein as ICU-COV(+) patients), COVID-19-negative critically ill patients (referred to herein as ICU-COV(−) patients), and healthy controls (HCs) without any diagnosed intestinal disorders (Figure 1).

The demographic and clinical characteristics of the participants are reported in Table 1. The three groups did not differ significantly concerning age or gender. The two groups of ICU patients were highly homogeneous with respect to the severity of the disease, as all of them required mechanical ventilation upon ICU admission. However, ICU-COV(−) patients enrolled before the pandemic differed significantly from the ICU-COV(+) patients with regard to known important risk factors for severe COVID-19, such as diabetes, chronic lung disease, and obesity [3]. These risk factors were taken into account as potential confounders in further association analyses. These two groups of patients also differed for reasons related to COVID-19 treatment, including a prone position, and dexamethasone and tocilizumab therapy. Of note, the C-reactive protein level (CRP), a marker of inflammation, was higher in ICU-COV(−) patients than in ICU-COV(+) patients upon admission (Table 1). The length of hospital stay before ICU admission was also significantly higher for ICU-COV(+) than for ICU-COV(−) patients (*p* < 0.0001), during which the former received more corticosteroid treatment (*p* < 0.0001, dexamethasone for ICU-COV(+) and methylprednisolone for ICU-COV(−)). However, antibiotic treatment before ICU admission was not significantly different between the two groups, although the type and doses of antibiotics administered could not be collected from the clinical history of the patients as most of these individuals were transferred to the ICU from other hospitals.

During the ICU stay, four (12.5%) ICU-COV(−) and six (11.3%) ICU-COV-(+) patients died. As hospital protocol, to eliminate or prevent possible bacterial co-infection with SARS-CoV2, all critically ill patients received, upon admission day, a prophylactic dose of an antibiotic such as beta-lactam alone (21/32, 66%), and the ICU-COV(+) patients received a combination of beta-lactam and macrolides (37/53, 70%), from the ICU admission day until the absence of urinary pneumococcal or Legionella antigen detection and a negative bacterial culture (online Appendix A). The latter combination therapy has been shown to significantly improve the prognosis of severe community-acquired pneumonia patients hospitalized in the ICU compared with a beta-lactam therapy not involving macrolides [14].

Positive correlations were observed between the clinical and laboratory data collected from ICU-COV(+) patients within 24 h of ICU admission: between the APACHE II (Acute Physiology and Chronic Health Evaluation II) score and age, between bacterial load and body mass index (BMI), between IgG and IgM levels, between troponin level and age, and between troponin and IL-6 level (online Appendix A). Negative correlations were observed between the PaO2/FiO2 ratio and BMI, between IgG and the lactate dehydrogenase level (LDH), and between IL-6 and CRP. A high APACHE II score reflects the severity of the disease for patients admitted to the ICU; a low PaO2/FiO2 ratio is an indicator of hypoxemia; and high levels of IL-6 and CRP (whose production is stimulated by IL-6), which are inflammatory markers, predict the need for mechanical ventilation. These correlations suggest that age and BMI, as expected, should be considered as risk factors for mechanical ventilation and that higher inflammation at ICU admission could increase the length of stay in this unit.

### 2.2. Sample Processing

The number of samples collected decreased over time, from T0 to T2W, due to 25 ICU discharges and 7 ICU deaths, and from T0 to T4W, due to 7 additional ICU discharges and 2 ICU deaths. Additionally, as reported in Figure 1, several samples failed to provide amplified microbial DNA, which prevented further sequencing processes. All 53 blood samples were processed without complications.

### 2.3. Alteration of the Gut Bacterial Microbiome of ICU Patients before Their ICU Stay

On the ICU admission day, ICU-COV(−) patients presented an altered gut microbiome compared with the HCs, with a significantly lower bacterial richness (Chao1 index on 16S rRNA sequences) and lower bacterial load (copy number of 16S rRNA gene via qPCR) (*p* < 0.001, online Appendix A). This altered microbiome before ICU admission could be explained by the antibiotics administered to these patients in the ward before admission (Table 1). However, bacterial richness was not significantly lower for ICU-COV(+) patients compared with HCs. Indeed, patients who received antibiotics but not corticosteroids presented lower bacterial richness compared with those who did not receive antibiotics but who were administered corticosteroids (*p* = 0.003).

In total, 25 covariates related to patient clinical/laboratory variables, time inward, treatment before ICU admission, and comorbidities were evaluated for their potential impact on the microbiome composition using the adonis2 test. Individually, pre-ICU treatments with corticosteroids and antibiotics were found to be the two most important covariates that could impact microbiome composition; however, none of the tests were significant after FDR correction (FDR > 0.3, online Appendix A), suggesting that these variables did not affect the global composition of the microbiome.

To assess the differences in microbial composition between groups, we used fixed-effect models implemented to take into account the influence of confounders in the MaAsLin2 tool. Thirty-two bacterial species were over-represented and 35 were under-represented in ICU-COV(−) patients compared with HCs (FDR < 0.05). Among the depleted species, 25 (71%) belonged to the Clostridiales order (Clostridiaceae, Lachnospiraceae, and Ruminococcaceae), whereas *Prevotella*, *Porphyromonas*, *Fusobacterium*, *Campylobacter*, *Staphylococcus*, and Enterobacteriaceae were among the enriched species (online Appendix A).

ICU-COV(+) patients did not show a significant difference in alpha-diversity, but an enrichment of 34 bacterial species, and a depletion of 61 bacterial species compared with HCs (MaAsLin2 method, FDR < 0.05, online Appendix A). All ICU patients presented a lower bacterial load compared to HCs. Interestingly, compared with the ICU-COV(−) patients (n = 46), ICU-COV(+) (n = 32) were enriched in 18 bacterial species, of which six were assigned to the Actinobacteria phylum and the *Enterococcus* genus, and three were assigned to the Proteobacteria phylum (*Haemophilus parainfluenzae*, *Campylobacter*, and Desulfovibrionaceae) (MaAsLin2 method, FDR < 0.05, online Appendix A). The differences between the two groups of patients could be attributed to the direct impact of SARS-CoV-2 and/or the antibiotic and corticosteroid treatment received before ICU admission. To remove the potential effect of pre-ICU treatment, we compared the rectal swabs of the ICU patients who did not receive antibiotics or corticosteroids (n = 19 for ICU-COV(−) and n = 14 for ICU-COV(+)) before ICU admission. Five bacterial species were enriched in ICU-COV(+) patients (*Collinsella aerofaciens*, and species assigned to *Campylobacter*, *Enterococcus*, *Fusobacterium*, and Actinomycetaceae), and one species assigned to *Dorea* was enriched in ICU-COV(−) individuals (Figure 2). These bacterial species in the gut samples could be directly associated with SARS-CoV-2 infection.

### 2.4. Progressive Changes in the ICU Gut and Respiratory Bacterial Microbiomes over Time

The bacterial load observed upon ICU admission increased significantly after 2 weeks in the ICU for ICU-COV(−) patients (*p* < 0.001, ANOVA test), and bacterial richness decreased significantly in both ICU-COV(−) and ICU-(COV(+) patients (*p* < 0.001, ANOVA test) (Figure 3a,b, Appendix A).

Over 2 and 4 weeks after ICU admission, ICU-COV(−) and ICU-COV(+) patients, respectively, underwent a progressive global change in their gut microbiome (PERMANOVA test, adonis2 function, *p* < 0.005) (Figure 3c). ICU-COV(+) patients, for which tracheal aspirate samples were collected, also showed changes in their respiratory microbiome composition (PERMANOVA test, adonis2 function, *p* < 0.005) (Figure 3c).

Next, we evaluated changes in the taxonomic composition of the microbiome over time spent in the ICU. The different bacterial species involved in the dysbiosis are listed in Figure 4 and online Appendix A, for ICU-COV(−)-RS, ICU-COV(+)-RS, and ICU-COV(+)-TA, respectively. In this list, an unknown Enterobacteriaceae was the most significantly enriched over time in the GI tract of ICU-COV(−) patients (Figure 4a). In the GI tract of ICU-COV(+) patients, *Enterococcus* and *Staphylococcus* increased, whereas a species from the *Prevotella* genus was the most depleted over time (Figure 4b). Moreover, in the tracheal aspirate specimens from the ICU-COV(+) patients, the most increased species belonged to the *Pseudomonas* genus (Figure 4c). In ICU-COV(+) patients, 126 microbial species were found to be significantly different between the gut and tracheal microbiome (FDR < 0.05, online Appendix A). This large difference in the taxonomic profiles explained the clear separation between the gut and tracheal samples in the clustering analysis based on UniFrac indexes (Figure 3c). Next, we investigated whether the observed dysbiosis was associated with the patients’ characteristics.

### 2.5. Association between the Bacterial Microbiome and Patients’ Characteristics

We assessed the association between clinical/laboratory data and gut microbiome data of ICU-COV(−) and ICU-COV(+) patients collected at baseline (n = 78, 2 rectal swab samples could not be amplified), taking into account confounders such as age, gender, BMI, comorbidities, and pre-ICU treatment. Significant associations were observed between bacterial species and age, length of stay in the ICU, PaO2/FiO2 ratio, and CRP level (Figure 5). Two pathobionts (Campylobacter and a species from the Actinomycetaceae family) were positively associated with a low PaO2/FiO2 ratio and two bacterial species assigned to the *Dorea* and *Blautia* genera, which have been reported to produce beneficial metabolites and thus have health-promoting functions, were associated with a high PaO2/FiO2 ratio. Six bacterial species were negatively associated with CRP blood levels.

We also studied the association between clinical/laboratory data and tracheal aspirate microbiome data of ICU-COV(+) patients collected at three time points. Significant associations were observed between various bacterial species and age, length of stay in the ICU, and length of dexamethasone treatment (Figure 6). Of note, among these associations, *Pseudomonas* was found to be positively correlated with the length of ICU stay (FDR = 0.03), and *Streptococcus* was negatively correlated.

### 2.6. Dysbiosis Score and Association with Disease Severity

A dysbiosis score, similar to that proposed by Lloyd-Price et al. [15] based on Bray–Curtis dissimilarities, was calculated for each of the gut samples using the UniFrac distances. In our study, this score was defined as the median weighted or unweighted UniFrac distance of any given sample with a reference set that was built with the 17 stool samples from HCs. This scoring system indicated a significant alteration of the global microbiome in critically ill patients before admission compared with HCs, but the scores of the two ICU groups were not significantly different (based on both weighted and unweighted UniFrac metrics). Dysbiosis appeared to decrease during the ICU stay for both ICU-COV(−) and ICU-COV(+) patients (Figure 7a). This decrease could be attributed to the administration of dexamethasone for a total of 10 days, as recommended in standard guidelines. However, this recovery did not reach healthy levels after 4 weeks in the ICU; again, this observation could be attributed to the continuous use of antibiotics and the withdrawal of dexamethasone treatment. We then performed an association analysis between this dysbiosis score and patients’ characteristics. Of note, a significant negative association was found between the PaO2/FiO2 ratio (calculated on ICU admission day) and the dysbiosis score (Figure 7b). The PaO2/FiO2 ratio, which is the ratio of arterial oxygen partial pressure to fractional inspired oxygen, is one of the main indicators of ARDS severity and the main criterion to transfer patients to the ICU. Therefore, dysbiosis could be considered as another risk factor for disease severity and another criterion for ICU admission. No dysbiosis score could be calculated for the TA samples, as these could not be obtained from the HCs for obvious ethical reasons.

### 2.7. Gut and Respiratory Fungal Microbiome

As the fungal community had been reported to be altered in hospitalized COVID-19 patients [11], we analyzed the fungal microbiome (ITS2 region sequencing) of the gut and tracheal aspirate samples. During the ICU stay, we observed an increase in fungal load, possibly as a result of antibiotic administration (online Appendix A).

Of the 281 gut and respiratory samples collected, 71 provided a positive PCR amplification for fungal taxonomic profiling. Interestingly, a significantly higher alpha-diversity was observed in ICU-COV(+) patients compared with HCs (online Appendix A). The gut mycobiome of ICU-COV(−) patients was depleted of *Saccharomyces* and enriched in *Candida albicans*, while ICU-COV(+) patients were enriched in species assigned to the Ascomycota phylum and species belonging to *Malassezia* and *Stereum* genera compared with HCs (online Appendix A).

Over time in the ICU, the gut microbiome of ICU-COV(+) patients was depleted of *Saccharomyces* and enriched in *Steccherinum ochraceum* and species assigned to the *Peniophora* genus, the Hypocreales order, and the Agaricomycetes class compared with baseline (online Appendix A). In ICU-COV(+) patients, tracheal aspirate samples were depleted of *Saccharomyces,* but enriched in *Beauveria*. Based on all of the samples obtained from ICU-COV(−) patients, a species from *Aureobasidium* was associated with mortality.

## 3. Discussion

The microbiome of the lower respiratory tract has received little attention in the context of SARS-CoV-2 infection, in particular in critically ill patients, in a longitudinal setting, and combining the gut bacterial and fungal microbiome with the systemic immune response analysis. In this study, the ICU-COV(+) cohort was enrolled between the second and fourth surge of COVID-19 cases in Catalonia (Spain), while the ICU-COV(−) cohort was recruited in the same ICU before the pandemic, which explained the differences in treatments, such as the use of corticosteroids and the type of antibiotics administered, which specifically targeted severe pneumonia with acute respiratory distress syndrome as a result of SARS-CoV-2 infection. Dexamethasone, a corticosteroid administered for 10 days, is recommended for patients hospitalized with COVID-19 who are receiving either invasive mechanical ventilation or oxygen alone [16,17]. Tocilizumab, an immunomodulatory drug, was combined with dexamethasone for patients with rapidly increasing oxygen needs. The combination of the two drugs was proven to reduce the mortality in these groups of patients. None of the participants of this study received the vaccine anti-COVID-19, which spared us from any possible effect of the vaccine on the microbiome composition.

One of the originalities of our study is the obtention of tracheal aspirate samples from ICU-COV(+) patients in a longitudinal setting and a highly homogeneous group of ICU patients in terms of disease severity, as they were all in need of mechanical ventilation. A study by Llorens-Rico et al. [18] characterized the microbiome of the upper and lower respiratory tracts of COVID-19 patients in the ICU and its association with viral load and patients’ characteristics. One of their key findings was that the heterogeneity in disease severity, leading to the use of different oxygenation methods, should be considered an important confounder for microbiome analysis, which justified, in our study, the inclusion of ICU patients all in need of mechanical ventilation.

One of the most remarkable findings of this study is the alteration of the gut microbiome (an increase of pathobionts and depletion of potentially beneficial microorganisms) before ICU admission, which could be attributed to the treatments received by some patients in wards before entering the ICU. Indeed, previous reports have shown that mortality was high when patients were admitted to the hospital for days or weeks before entering the ICU [19]. The dysbiosis score obtained upon ICU admission day could be considered an additional indicator of disease severity and improve mortality predictions. A bacterial signature (six species) that discriminates between ICU-COV(−) and ICU-COV(+) patients independent of treatment received before ICU admission has not been reported previously, to the best of our knowledge, and could be a biomarker of COVID-19 infection. However, these findings should be validated with a much larger cohort.

These alterations worsened during the ICU stay, with a decrease in bacterial diversity and an increase in bacterial and fungal loads. These changes over time were probably related to the prophylactic doses of broad-spectrum antibiotics received by both groups of patients, and also to the corticoids administered to ICU-COV(+) patients. Moreover, the decreased diversity and increased load were associated with a reduction in beneficial bacteria and fungi, and an increase in pathobionts in the gut and the respiratory microbiomes. These changes may increase the length of stay in the ICU or the severity of ARDS. Indeed, bacterial species belonging to the Clostridiales order have been recognized as producers of butyrate, an end-product of fermentation, and thus are critical to maintaining gut homeostasis [20,21].

The findings on the demographic and clinical/laboratory data of the two ICU cohorts confirmed previous studies that found that age and obesity may contribute to the severity of the disease, as assessed by the PaO2/FiO2 ratio, the APACHE II index, and length of ICU stay [22]. They also validate previous studies showing that a marker of heart injury such as troponin is associated with age and inflammation (IL-6) [23]. One of the two bacterial species found to be negatively correlated with the PaO2/FiO2 ratio belongs to the Actinomycetaceae family, which indicates that it was associated with severe COVID-19 cases. This result is in line with a previous study that showed a positive correlation between *Actinomyces viscosus* (belonging to the Actinomycetaceae family) with COVID-19 severity [11]. Actinomyces spp. is known to cause pulmonary actinomycosis, a bacterial lung infection [24], and thus could emerge as a potential trigger or perpetuator of disease severity in ICU patients

The lower respiratory tract has been shown to carry a complex bacterial and fungal microbiome [25]. This microbial community was also altered before ICU admission and was significantly impacted during ICU stay. Regarding the gut samples, the dynamic of the tracheal aspirate samples was characterized by the depletion of several beneficial bacteria, including *Streptococcus,* and the enrichment of *Pseudomonas*. The latter is one of the most prevalent pathogens in ICU infections [26,27]. Here, we demonstrate that a high-throughput sequencing approach implemented in the ICU for pathogen detection, as well as the follow-up of commensal microorganisms, can be used as a complementary or even as an alternative method to traditional isolation techniques.

Despite the considerable effort channeled into recruitment and sampling, and a comprehensive examination of the data, this study has several limitations. The SARS-CoV-2 load in fecal or tracheal aspirate samples was not examined to assess an association between bacterial, fungal, and viral loads. Indeed, the detection of viruses would require the collection of an additional aliquot for each sample type in an adequate solution to preserve the virus, and then a well-equipped laboratory to process the samples. This procedure was not approved by our local ethics committee due to a lack of knowledge about the transmission route of the virus at the beginning of the pandemic. Another limitation of our study is the use of 16S rRNA sequencing instead of shotgun sequencing, which could have provided a more refined microbiome taxonomic and functional profiling. The dysbiosis scores could not be calculated for respiratory samples due to the lack of appropriate controls (TA samples from healthy subjects), or for the fungal microbiome due to the low detection of fungi in human biological specimens. The comparison between fecal samples and rectal samples should be taken with caution, although many studies have shown that rectal swab samples reliably replicate the fecal microbial composition, as well as alpha- and beta-diversity [28,29]. Finally, the sample size is indeed relatively small and our findings should be validated using another cohort. However, considering the type of patients (critically ill with and without COVID-19), all in need of mechanical ventilation, and the fact that they were recruited during the second wave of the pandemic, during which the anti-COVID-19 vaccine was not yet available, we believe that our study is unique and does not present a lower number of patients (n = 76 critically patients) compared with other recent papers. Indeed, Llorens-Rico et al. recently published with n = 58 and with very heterogeneous critically ill patients [18]. Moreover, although Sulaiman et al. [13] recruited a higher number of critically ill patients (n = 142), they performed only a cross-sectional study. In our study, we were able to recruit a very homogeneous cohort of critically ill patients and collect tracheal aspirate samples for up to three time points, which allowed for an appropriate evaluation of the microbiome dynamic.

## 4. Methods and Materials

### 4.1. Study Design

This study is observational and includes three groups of participants. ICU-COV(+) patients (n = 53) were recruited between November 2020 and April 2021. Rectal swabs (RSs), tracheal aspirates (TAs), and blood were collected from these patients at three time points: within 24 h of ICU admission, and 2 and 4 weeks after admission, unless they were discharged or deceased. ICU-COV(−) patients (n = 32) were recruited before the pandemic between March 2017 and January 2018. Rectal swabs and clinical and laboratory data were collected from these patients upon ICU admission, and 1 and 2 weeks after admission. A final group consisting of COV(−) HCs (n = 17) did not present any diagnosed diseases and were recruited between May 2017 and August 2020. For these subjects, stool samples were collected at baseline and 4 weeks later. All of the participants were over 18 years old, none of them received the COVID-19 vaccine, and all ICU patients required mechanical ventilation upon admission to the ICU.

### 4.2. Sample Collection

For bacterial and fungal microbiome analysis, 34 fecal samples (FCs) were collected from HCs from a previous study [30], 83 rectal swabs from ICU-COV(−), and 86 RSs and 78 TAs from ICU-COV(+) patients. A total of 281 gut and respiratory samples were collected for bacterial and fungal microbiome analysis and 53 blood samples were obtained from ICU-COV(+) patients for measuring the levels of IgM, IgG, and IL-6. All of the samples were stored at −80 °C the hour after collection and were shipped to the Microbiome Lab for further processing using dry ice.

### 4.3. Microbial DNA Extraction

FCs (250–300 mg), RSs, and TAs (250–300 mL) were processed for genomic DNA extraction, as previously described [31,32] and following the recommendations from the International Human Microbiome Standards [33] in the Microbiome Lab of the Vall d’Hebron Research Institute (Barcelona, Spain). Extracted genomic DNA was resuspended with 200 μL of Tris-EDTA (TE) for FCs and with 50 µL of TE for RCs and TAs. As RSs and TAs were considered medium-low and low-bacterial load samples, respectively, additional controls during PCR and sequencing were added to control for possible contamination during extraction and PCR amplification. All amplicons with possible contamination were discarded and repeated.

### 4.4. Microbiome Composition Analysis

For prokaryotic taxonomy profiling, the V4 hypervariable region of the 16S rRNA gene was used [32]. Sequencing was performed using the MiSeq Illumina platform. Sequences were analyzed using QIIME2 [34]. The sequences were demultiplexed to assign reads to the appropriate samples and were then denoised and dereplicated into amplicon sequence variants (ASVs) using the dada2 tool, which also filtered out chimeras. Each sequence read was trimmed to 298 bp. A total of 5.4 million sequences of the 16S rRNA gene were generated from the 232 amplifiable samples, with a mean of 22,868 sequences per sample. A feature table was generated for all samples, with a minimum of 4300 sequences per sample. The feature table was then used to perform taxonomic classification, alpha- and beta-diversity analyses, and differential abundance measurements in the different experimental groups. For bacterial profiling, taxonomy was assigned to each ASV using the 16S Greengenes database, the gg_13_8_99 release, which contains 202,421 bacterial and archaeal sequences. Downstream analysis was performed at the species level.

For fungal taxonomic profiling, the ITS2 region was PCR-amplified. The following primers were used for this purpose: fITS7 = 5′ NNNNNNNNGTGARTCATCGAATCTTTG 3′ and ITS4 = 5′ NNNNNNNNTCCTCCGCTTATTGATATGC 3′. Sequence analysis was performed with a similar approach as for 16S rRNA sequences using the QIIME2 pipeline. Each sequence read was trimmed to 284 bp. A total of 2.5 million sequences of the ITS2 region were generated from the 71 amplifiable samples, with a mean of 32,000 sequences per sample. A feature table was generated for all samples, with a minimum of 1535 sequences per sample. Taxonomy was assigned to each ASV using a combined and non-redundant UniRef-RefSeq database, which contains 96,388 fungal sequences.

### 4.5. Microbiome Load Analysis

Bacterial and fungal loads were estimated in FCs, RSs, and TAs by targeting the V4 region of the 16S rRNA gene and the ITS2 region, respectively. Amplification was done using a 7500 Fast real-time PCR system (Applied Biosystems, Foster City, CA, USA). The fungal ITS2 region was amplified using ITS2-fungi-sense (5′-GTG ART CAT CGA ATC TTT-3′) and ITS2-fungi-antisense (5′-GAT ATG CTT AAG TTC AGC GGG T-3′) primers. The V4 region of the 16S rRNA genes (290 bp) was amplified using the following primers: V4F_517_17 (5′-GCC AGC AGC CGC GGT AA-3′) and V4R_ 805_19 (5′-GAC TAC CAG GGT ATC TAA T-3′).

qPCR was carried out in a final volume of 25 μL using Power SYBR green PCR master mix (Fisher Scientific, Madrid, Spain). The qPCR reaction conditions were 50 °C for 2 min, 95 °C for 10 min, followed by 38 cycles at 95 °C for 15 s and 60 °C for 1 min. In the case of the ITS2 region, the conditions were 50 °C for 2 min, 95 °C for 2 min, followed by 40 cycles at 95 °C for 30 s, 55 °C for 30 s, and 72 °C for 60 s, and a final extension cycle of 72 °C for 10 min. Genomic DNA samples were diluted 100-fold and performed per duplicate and triplicate, respectively. For further analysis, the mean values were considered. Melting curves were inspected after amplification to evaluate the specificity of the qPCR. To generate standard curves, calculated amounts of linearized plasmids, in which the amplified region from a control bacterium had been inserted, were used. The plasmid concentration was measured using a NanoDrop ND-1000 spectrophotometer (Thermo Scientific, Wilmington, DE, USA), and the number of copies was calculated from the molecular weight of the plasmid. Serial dilutions of the template DNA were amplified to extrapolate bacterial (from 10^2^ to 10^7^) and fungal (from 10^0^ to 10^6^) copy numbers. The results were expressed in copies per ng of genomic DNA extracted.

### 4.6. Plasma Immunoglobulin and Cytokine Measurements

Details on these measurements can be found in Appendix A.

### 4.7. Statistical Analysis

The statistical analysis was performed using QIIME2 (v2021.2.0), Prism (v8.4.3), and R Studio (v2021.09.1). The Kolmogorov–Smirnov and the Shapiro–Wilk tests were used to evaluate the normality of the data. Two-sided Mann–Whitney U or Wilcoxon tests were performed to compare two groups, and PERMANOVA (permutational multivariate analysis of variance) or Kruskal–Wallis tests were used for more than two groups. To study the association between the microbiome data and clinical and laboratory variables, we used linear mixed models as implemented in the Microbiome Multivariable Association with Linear Models (MaAsLin2) package [35]. MaAsLin2 was set up with the following parameters: normalization = “TSS”, transform = “LOG”, correction = “BH”, analysis_method = “LM”, max_significance = 0.25 (default significance threshold), min_abundance = 20, min_prevalence = 0.1. The age, gender, and BMI of the participants were added as fixed effects. When time points were used as fixed effects, “baseline or T0” was used as a reference. Results with a false-discovery rate (FDR) lower than 0.05 were considered significant.

## 5. Conclusions

Despite the mentioned limitations, this study provides new insights into the characterization of the dynamics of the bacterial and fungal microbiome of the gastrointestinal and respiratory tracts, two key body sites for SARS-CoV-2 multiplication. The study revealed that critically ill patients with COVID-19 were associated with a bacterial signature upon the ICU admission day, which could be used as a biomarker of SARS-CoV-2 infection. The severity of COVID-19, along with the specific treatment administered to critically ill patients, worsened the dysbiosis of their gut and respiratory microbiomes. As a more immediate application, we propose the use of a high-throughput sequencing approach as an alternative to traditional isolation techniques to monitor ICU patient infection and health status.

## Figures and Tables

**Figure 1 ijms-23-15808-f001:**
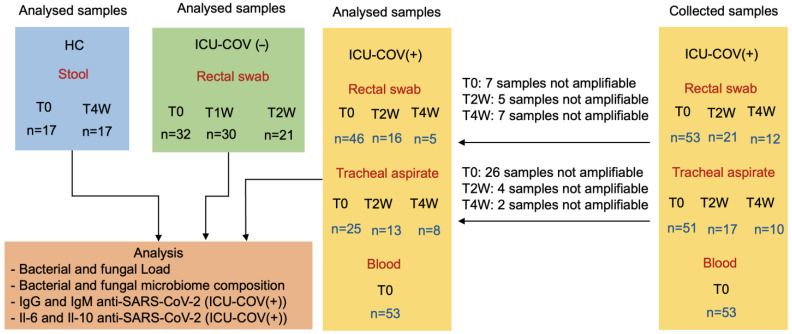
Diagram of participant groups and number of samples collected and processed. The study population consisted of three groups of subjects: 17 HCs, 32 patients admitted to the ICU before the SARS-CoV-2 pandemic (ICU-COV(−)), and 53 patients with COVID-19 admitted to the ICU (ICU-COV(+)). Fecal samples were collected from HCs, rectal swabs from all ICU patients, and tracheal aspirates and blood from ICU-COV(+) patients. HCs and ICU-COV(+) patients were monitored for up to 4 weeks and ICU-COV(−) patients were monitored for up to 2 weeks. The reason the number of samples decreased over time was that the patients were either discharged or deceased before their samples were collected, and the reason the number of samples analyzed is lower than those collected was that some samples were not amplifiable by PCR.

**Figure 2 ijms-23-15808-f002:**
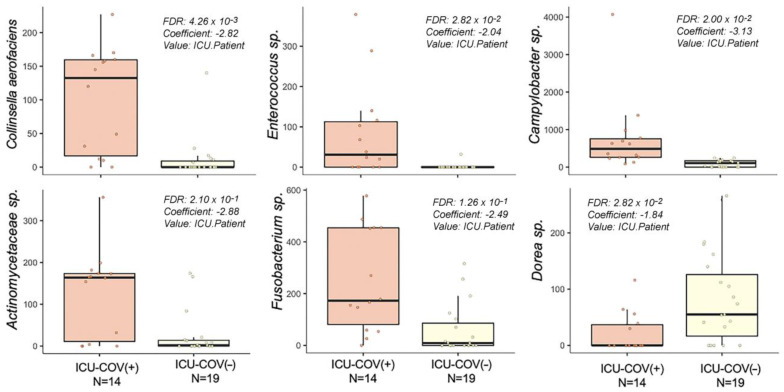
Gut bacterial taxa associated with ICU-COV(+) patients and their predictive value. Comparison of the gut microbiome sequence data between ICU-COV(+) and ICU-COV(−) patients who did not receive antibiotics or corticosteroids before ICU admission using MaAsLin2 identified six bacterial species (FDR < 0.22). ICU = intensive care unit; COV(−) = COVID-19-negative; COV(+) = COVID-19-positive; RS = rectal swab; TA = tracheal aspirate.

**Figure 3 ijms-23-15808-f003:**
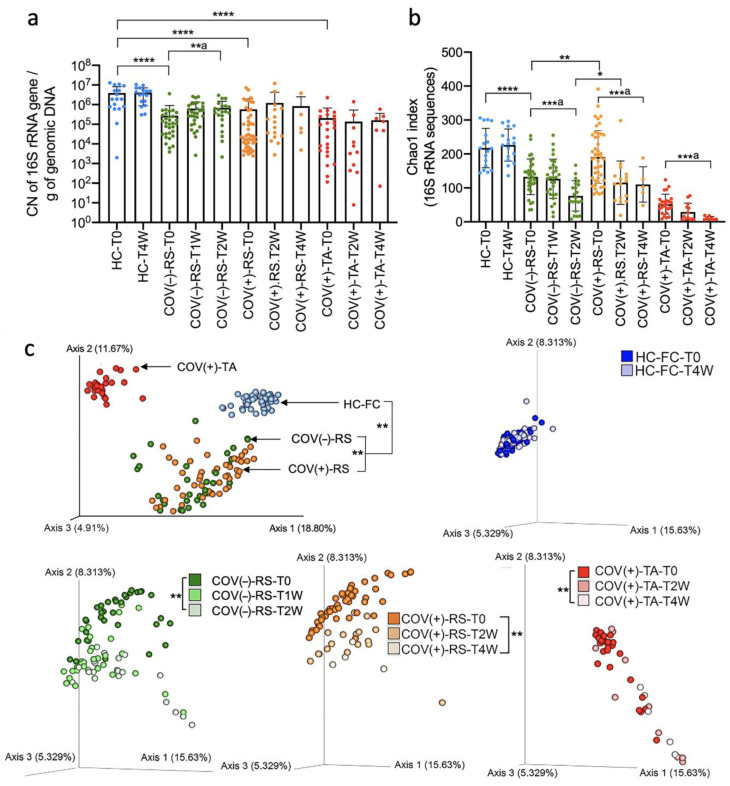
Dynamics of the gut and respiratory bacterial load and diversity. (**a**) Bacterial load in critically ill patients as assessed by qPCR of the V4 region of the 16S rRNA gene at ICU admission and over time spent in the ICU. * *p* < 0.05, ** *p* < 0.001, *** *p* < 0.0001, **** *p* < 0.0001, Mann–Whitney test unless three groups were tested, then Kruskal–Wallis test was applied (**a**). (**b**) Bacterial richness as assessed by the Chao1 index analysis of the 16S rRNA sequences upon ICU admission, and progressive decrease in this richness in both the gut and respiratory microbiome from admission day to 2 weeks for ICU-COV(−) patients, up to 4 weeks for ICU-COV(+) patients, one fecal sample at baseline and one 4 weeks later for HCs. a. Kruskal–Wallis test and ANOVA test for more than two groups and non-parametric or parametric data, respectively. (**c**) Beta-diversity analyses using unweighted UniFrac Principal Coordinate Analysis (PCoA). T0 = baseline; T1W = sample collected 1 week after baseline; T2W = sample collected after 2 weeks after baseline; T4W = samples collected after 4 weeks after baseline. ** *p* < 0.005, 5 PERMANOVA test (adonis2 function). ICU = intensive care unit; COV(−) = COVID-19-negative; COV(+) = COVID-19-positive; RS = rectal swab; TA = tracheal aspirate.

**Figure 4 ijms-23-15808-f004:**
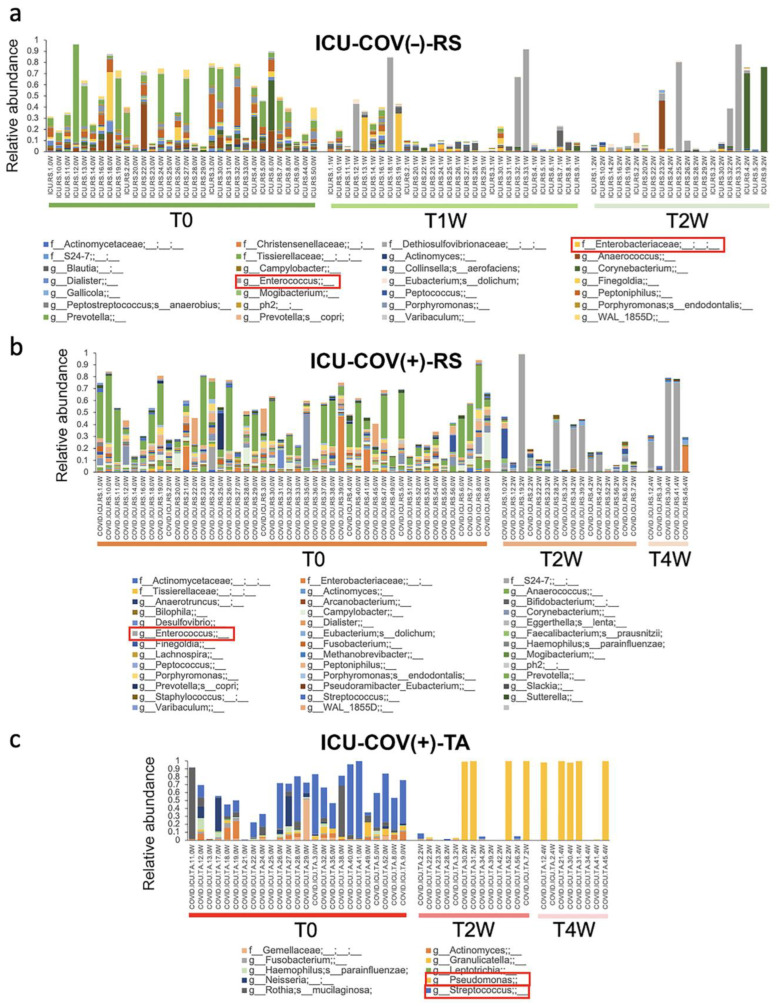
Bacterial taxonomic profiling over time in the ICU. (**a**) Significant changes in the gut microbiome of the ICU-COV(−) patients at baseline, and 1 and 2 weeks after ICU admission. (**b**) Significant changes in the gut microbiome of ICU-COV(+) patients at baseline, and 2 and 4 weeks after admission. (**c**) Significant changes in the tracheal aspirates of ICU-COV(+) patients at baseline, and 2 and 4 weeks after ICU admission. Intra-group variability across the three time points was evaluated using MaAsLin2. Only significant results have been plotted (FDR < 0.05). ICU = intensive care unit; COV(−) = COVID-19-negative; COV(+) = COVID-19-positive; RS = rectal swab; TA = tracheal aspirate. Species inside most affected over time are highlighted in red boxes.

**Figure 5 ijms-23-15808-f005:**
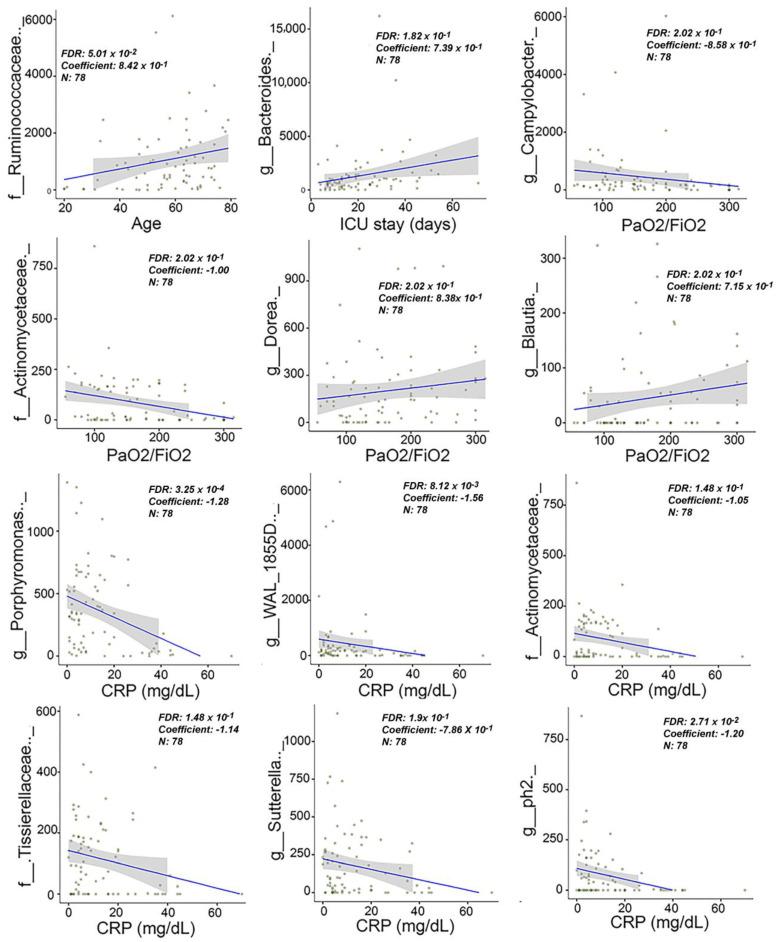
Association between clinical and laboratory data and gut microbiome composition data at baseline using MaAsLin2. Only significant results are shown (FDR < 0.25 are specified for each association). Gut microbiome data are plotted in raw counts.

**Figure 6 ijms-23-15808-f006:**
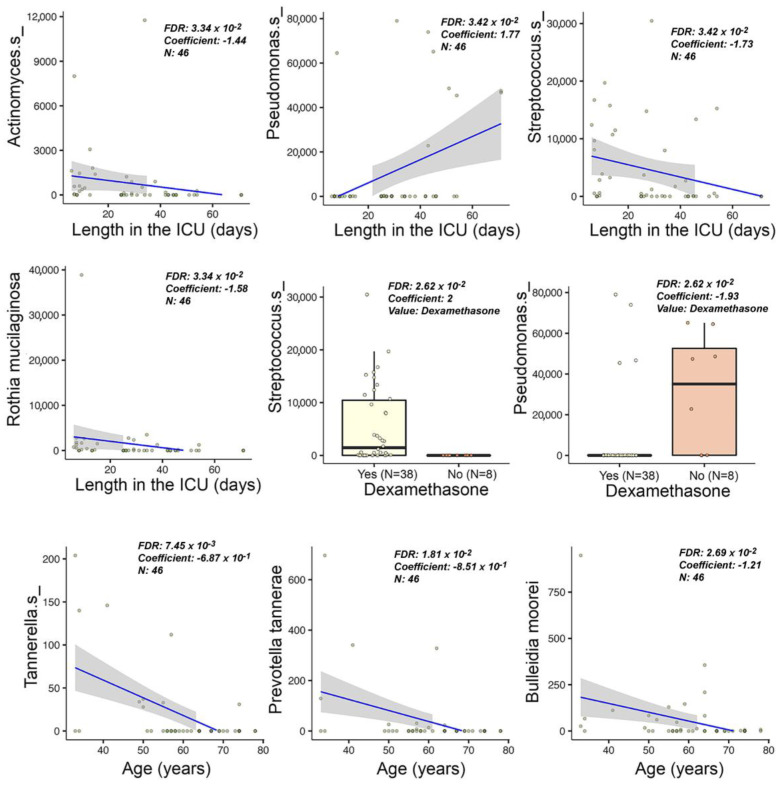
Association between clinical/laboratory data and microbiome composition data of the tracheal aspirate at baseline for ICU-COV(+) patients using MaAsLin2. Only the most significant results are shown (FDR < 0.05 are specified for each association). Respiratory microbiome data are plotted in raw counts.

**Figure 7 ijms-23-15808-f007:**
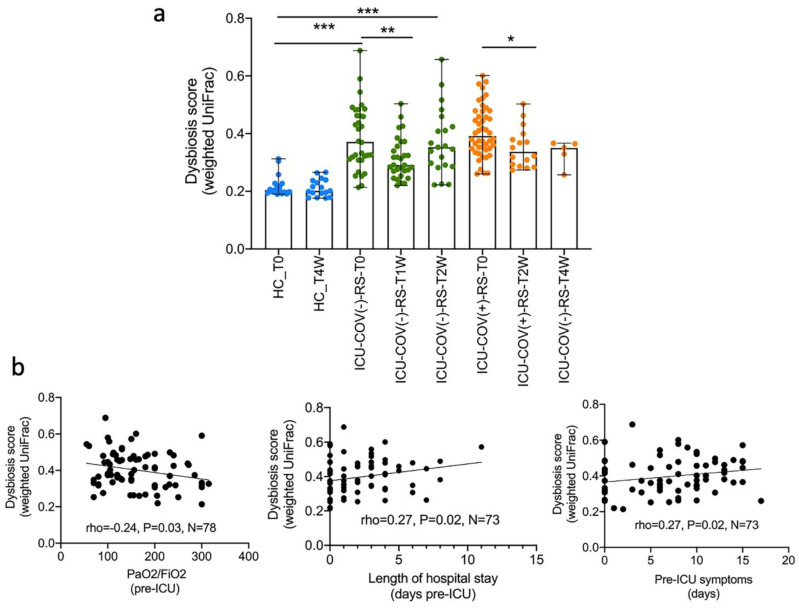
Dysbiosis scores calculated for each sample and association with clinical data. (**a**) Dysbiosis scores in critically ill patients were calculated based on the weighted UniFrac distance metrics of the 16S rRNA sequence data of rectal swab samples, using fecal samples of HCs as reference data. * *p* < 0.05, ** *p* < 0.001, *** *p* < 0.0001, Mann–Whitney test. (**b**) Correlation between dysbiosis scores and clinical data such as the PaO2/FiO2 ratio, length of ward stay before ICU, and duration of pre-ICU clinical symptoms (Spearman correlation test). The correlations were performed using rectal swab samples obtained at baseline for both ICU groups.

**Table 1 ijms-23-15808-t001:** Participant’s characteristics.

Column1	HC	ICU-COV(−)	ICU COV(+)	Statistics
**Baseline demographic characteristics**	n = 17	n = 32	n = 46	
Age (median, years)	55	58	61	*p* = 0.12 (Kruskal–Wallis)
Female (%)	9 (52.3%)	12 (37.5%)	14 (30%)	*p* = 0.25 (Chi-square)
Body mass index (kg/m^2^) > 30	1 (5.9%)	7 (21.8%)	18 (39.1%)	***p* = 0.02 (Chi-square)**
**Comorbidities and other underlined conditions**				
Arterial hypertension	ND	12 (37.5%)	25 (54.3%)	*p* = 0.17 (Fisher’s)
Chronic obstructive pulmonary disease (COPD)	ND	1 (3.1%)	12 (26%)	***p* = 0.01 (Fisher’s)**
Diabetes mellitus (type 2)	ND	18 (56.2%)	16 (34.8%)	*p* = 0.068 (Fisher’s)
Immunosuppression	ND	0	12 (26%)	***p* = 0.0010 (Fisher’s)**
**Index/Scores**				
Acute respiratory distress syndrome (ARDS)	ND	ND	35 (76%)	
PaO2/FiO2 < 150	ND	ND	31 (67.4%)	
**Therapies**	ND			
Anticoagulant prophylaxis	ND	32 (100%)	46 (100%)	
Dexamethasone	ND	0	46 (100%)	
Neuromuscular-blocking drug	ND	12 (19.3%)	40 (86.9%)	***p* < 0.0001 (Fisher’s)**
Orotracheal intubation	ND	32 (100%)	46 (100%)	
Prone position	ND	6 (18.7%)	34 (73.9%)	***p* < 0.0001 (Fisher’s)**
Remdesivir	ND	0	2 (4.3%)	*p* = 0.5 (Fisher’s)
Tocilizumab	ND	0	9 (19.5%)	***p* = 0.0088 (Fisher’s)**
Tracheotomy	ND	13 (40.6%)	12 (26%)	*p* = 0.61 (Fisher’s)
**Therapy pre-ICU**				
Pre-ICU antibiotics, mean days (SD)	0	0.7 (1.18)	1 (1.7)	*p* = 0.61 (Mann–Whitney)
Pre-ICU corticosteroids, mean days (SD)	0	0.1 (0.5)	2.1 (2.1)	***p* < 0.0001 (Mann–Whitney)**
**Timing**				
Total length of hospitalization, mean days (SD)	ND	30.91	31.09	*p* = 0.32 (Mann–Whitney)
Length of ICU stay, mean days (SD)	ND	20.63	21.91	*p* = 0.43 (Mann–Whitney)
Pre-ICU length of hospital stay, mean days (SD)	0	0.5 (0.96)	9.7 (3.3)	***p* < 0.0001 (Mann–Whitney)**
Pre-ICU symptoms, mean days (SD)	0	2.35 (4)	3 (2.5)	***p* < 0.0001 (Mann–Whitney)**
**Laboratory data at admission**				
C-reactive protein (CRP) (mg/dL)	ND	19.1	9.6	***p* = 0.03 (Mann–Whitney)**
D-Dimer (ng/mL)	ND	ND	4825	
Ferritin (ng/mL)	ND	ND	1195	
Lactate Dehydrogenase (LDH) (U/L)	ND	ND	459	
Leukopenia (10*3/mcL)	ND	ND	12	
Troponin (pg/mL)	ND	ND	13.66	

Significant differences are shown in bold.

## Data Availability

Data are available in a public, open-access repository. http://www.ncbi.nlm.nih.gov/bioproject/814397 (accessed on 1 November 2022). Raw sequence data are available in the Sequence Read Archive (SRA) under BioProject accession PRJNA814397.

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
