# Peer review of "Dysbiosis: An Indicator of COVID-19 Severity in Critically Ill Patients"

_ijms, 2022, doi:10.3390/ijms232415808_

Round 1

Reviewer 1 Report

This is a very interesting, well performed and well described study. I have no suggestions for corrections or changes to the text.

Author Response

Dear reviewer,

We appreciate your positive feedback and acknowledge the effort in reviewing our manuscript.

Best regards

Reviewer 2 Report

In the study by Cuenca and Soler et al. , COVID-19 patients requiring mechanical ventilation in an ICU were compared with patients without COVID-19 in the same situation and their microbiomes and fungal taxonomic profile were examined in more detail. In addition, the data were compared with healthy controls. The authors conclude from their research that gut and respiratory microbiome dysbiosis and bacterial signatures associated with critical illness emerge as biomarkers of COVID-19 severity and a potential predictor of ICU length of stay.

The manuscript is well written, however, the manuscript has some limitations (also mostly listed by the authors themselves) which weaken the manuscript and make it difficult to interpret the results. The biggest issue are the large differences between the three groups (HC, COV+, COV-) at different levels. The patients were recruited at different time periods (the COV negative patients were recruited before the Covid pandemic). What might also be an issue is the different collection of the microbiota samples. In the ICU patients rectal swabs were used, the healthy controls gave stool samples. In order to ensure comparability and prevent potential bias, the samples should be stored and collected in the same way and from the same site, otherwise differences found might also due to the inhomogeneity in collecting samples. Participants also differed in their body weight (mean body weight/ BMI is missing) which might also affect the microbiota outcome. Moreover, the difference in pre-ICU length of hospital stay might have an influence here too due to the “hospital-associated microbiota” (PMID: 28539477)

Author Response

Dear reviewer,

We appreciate the reviewer’s report. One of the main reasons the ICU-COV(-) patients could not be recruited in the same window period as the ICU-COV(+) patients was that, during the first and second wave of the pandemic, during which we started the study, our ICU department was converted into an exclusive severe COVID patients unit. Therefore, the recruitment of ICU-COV(-) patients simultaneously with the ICU-COV(+) patients would have not been possible. However, we included in the study only ICU-COV(-) patients in need of mechanical ventilation to match with the COVID-19 patients.

For a similar reason, the recruitment of healthy volunteers to collect swab samples, to match with the ICU patients’ samples,  would not have been realistically feasible during this period of the pandemic. Indeed, in Spain, after the period of lockdown, it was not recommended for more than one year for healthy people to come to the hospital, where our Microbiome Lab is located. Therefore, we could not have collected swab samples from healthy individuals for this study.

However, in order to take into account the differences in body weight and pre-ICU length of hospital stay between the different groups of participants, we added these variables as a fixed effect in the MaAsLin2 tool (statistical method for association analysis of microbial community meta-omics profiles) (see method section).

Reviewer 3 Report

The manuscript reports on the comprehensive analysis of gut and bronchoalveolar microbiomes in patients with severe COVID-19 admitted to the intensive care unit. This theme is of the great research and practice-oriented interest. The manuscript addresses the crucial question of severe COVID-19, namely additional signs of severity, prevention and treatment of severe patients with COVID-19. The authors used modern methods and techniques, including blood analyses, NGS of rectal swabs and bronchoalveolar lavage. Control cohorts of patients were studied with a similar state but without COVID-19 as well as healthy persons. Contemporary bioinformatic and statistical methods were used for this study. The study is original and seems to be reproducible. The article is written in clear and professional manner, but some inconsistencies need proper corrections.

 Major corrections:

1. An abstract does not present the most considerable results of the study. For this, the main results should be strictly separated from additional ones. Particularly, the most significant differences in microbiomes of patients with COVID-19 against control persons are not given, including drastic decrease of taxonomic richness and valuable gut bacterial taxa associated with ICU-COV(+) patients. Also, outstanding findings of the temporal dynamics in quantitative and qualitative microbiome features should be presented in the abstract. But gut and bronchoalveolar microbiomes would be better to be described separately from each other.

2. Part 2.6, Figure 7, and corresponding parts of other text describe “dysbiosis score” evaluated as a function of UniFrac distances between healthy and critically ill patients. However, the dynamics of the microbiomes composition based on the dysbiosis score (Fig. 7) does not meet the dynamics of the same microbiomes described by use of a bacterial load and Chao1 index (Fig. 3). Taking into account that currently decrease of biodiversity is considered the main criterion of dysbiosis, and that it has been recorded in the gut microbiomes of patients with COVID-19 in this study, all the mentioned facts need an explanation. In the light of this, the authors’ interpretation of the facts (Lines 261-265) sounds contradictory and unreliable. Indeed, “decrease of dysbiosis during the ICU stay due to administration of dexamethasone” does not agree with a sharp decrease of Chao1 index (Fig. 3) demonstrating the deep dysbiotic disorders. Decrease of Chao1 index in the bronchoalveolar samples (Fig. 3) seems even more dramatic than in the gut ones. So, in this situation a weak reliability of the dysbiosis score used should be noted here. There are some alternatives might be proposed to the authors. First, the authors’ formula of dysbiotic disorders could be improved based on the available literature or their own development. Second, the dysbiotic disorders in the patients observed may be described as a complex of qualitative and quantitative changes in microbiomes, including diversity indices, bacterial load, and differential abundance of bacterial taxa. Really it would be more compliant with the current conception of dysbiosis. In addition, I recommend to try Simpson’s index as another score reflecting community evenness. It usually confirms well the drastic qualitative changes accompanied with predominance of one or few bacterial taxa in a microbiome.

3. Discussion covers main findings and results, but does not contain any speculations about role of those bacterial taxa, which have been demonstrated to feature a differential abundance at the statistically significant level except for Streptococcus and Pseudomonas (Line 355). Even a mention of Clostridiales and their beneficial role (Lines 342-345) seems unrelated to the data obtained. Correlations with microbiota in patients with severe COVID-19 are also absent. However, such fragments can improve understanding of the changes in microbiomes recorded in this study and attract more specialists in COVID-19 microecology.

 Minor corrections:

1. Line 108-110. It is unclear, why the antibiotics were prescribed “until the absence of urinary pneumococcal or Legionella antigen detection and a negative bacterial culture”, as though the patients were suffered from legionnaires’ disease or pneumococcal infection or septicemia? Moreover, in Line 106 “a prophylactic dose” of antibiotics is mentioned. This unclarity should be resolved.

2. Line 118. Abbreviations BMI should be deciphered at first mention.

3. Line 147 and below. If the PCR-based bacterial load is interpreted, it never is applied for description of microbiome composition, because its quantitative sense, whereas microbiome composition is a qualitative term. This unclarity should be resolved.

4. Line 153 and below. Term “bacterial species” as well as other taxa may be used only if the statistical analyses are performed at different taxonomic levels, including species etc. At the same time, all the supplementary tables look as though statistical analysis was performed at ASV level only, without their clustering into species, genera and other taxa. If there were several analyses after ASVs combining into taxa, at every taxonomic level, it should be indicated clearly in the M&M section, as well as in all captions for figures and tables, where it is necessary.

5. Figure 3. Percentage numbers at all axes in all PCoA plots should be seen clearly. Now, they are absent at axes 3 in the top left and lower right plots.

6. Line 206. Unknown Enterobacteriaceae seems curious, taking into account that all main members of this family have been described by the current moment. Possibly, it may be determined by a poor discrimination of the 16S Greengenes database, which last release was in 1999. For such poorly identified ASVs other databases are recommended, such as RDP or Genbank (NCBI), where simple BLAST can substantially specify taxonomy of an OTE or ASV.

7. Line 243. “Detection of multi-drug resistant bacteria”, which are not mentioned elsewhere in the manuscript. This unclarity should be resolved.

8. Lines 398-400 and 408-410 contain absolutely identical statement “All participants…”. This unclarity should be resolved.

9. Line 419. “All PCR with possible contamination was…” should be exchanged by “All amplicons with possible contamination were…”

Author Response

Dear reviewer,

We appreciate your time for commenting on our work. Those comments are all valuable and very helpful for revising and improving our paper. We have studied the comments carefully and have made corrections, which we hope will fit your requests. Our answers are added in blue below your comments. I'm looking forward to hearing from you soon.

The manuscript reports on the comprehensive analysis of gut and bronchoalveolar microbiomes in patients with severe COVID-19 admitted to the intensive care unit. This theme is of the great research and practice-oriented interest. The manuscript addresses the crucial question of severe COVID-19, namely additional signs of severity, prevention and treatment of severe patients with COVID-19. The authors used modern methods and techniques, including blood analyses, NGS of rectal swabs and bronchoalveolar lavage. Control cohorts of patients were studied with a similar state but without COVID-19 as well as healthy persons. Contemporary bioinformatic and statistical methods were used for this study. The study is original and seems to be reproducible. The article is written in clear and professional manner, but some inconsistencies need proper corrections.

Major corrections:

1. An abstract does not present the most considerable results of the study. For this, the main results should be strictly separated from additional ones. Particularly, the most significant differences in microbiomes of patients with COVID-19 against control persons are not given, including drastic decrease of taxonomic richness and valuable gut bacterial taxa associated with ICU-COV(+) patients. Also, outstanding findings of the temporal dynamics in quantitative and qualitative microbiome features should be presented in the abstract. But gut and bronchoalveolar microbiomes would be better to be described separately from each other.

Response:

We thank the reviewer for his/her help to improve the abstract. As suggested, we have made the following changes:

“…Here we examined the dynamics of the gut and respiratory microbiomes in severe COVID-19 cases in need of mechanical ventilation in the intensive care unit (ICU). We recruited 85 critically ill patients (53 with COVID-19 and 32 without COVID-19) and 17 healthy controls (HCs) and monitored them for up to 4 weeks. We analyzed the bacterial and fungal taxonomic profiles and loads of 232 gut and respiratory samples. We also measured blood levels of Interleukin 6, IgG, and IgM in COVID-19 patients. At ICU admission, bacterial composition and load in the gut and respiratory samples were altered in critically ill patients compared to HCs. During their ICU stay, patients experienced increased bacterial and fungal loads, drastic decreased bacterial richness, and progressive changes in bacterial and fungal taxonomic profiles. In gut samples, six bacterial taxa could discriminate ICU-COV(+) from ICU-COV(-) cases at ICU admission and bacterial taxa were associated with age, PaO2/FiO2, and CRP levels. In respiratory samples ICU-COV(+) patients, bacterial signatures including Pseudomonas and Streptococcus were found correlated with the length of the ICU stay.  Our findings demonstrated that gut and respiratory microbiome dysbiosis and bacterial signatures associated with critical illness emerge as biomarkers of COVID-19 severity and a potential predictor of ICU length of stay. We propose using a high-throughput sequencing approach as an alternative to traditional isolation techniques to monitor ICU patient infection…”

2. Part 2.6, Figure 7, and corresponding parts of other text describe “dysbiosis score” evaluated as a function of UniFrac distances between healthy and critically ill patients. However, the dynamics of the microbiomes composition based on the dysbiosis score (Fig. 7) does not meet the dynamics of the same microbiomes described by use of a bacterial load and Chao1 index (Fig. 3). Taking into account that currently decrease of biodiversity is considered the main criterion of dysbiosis, and that it has been recorded in the gut microbiomes of patients with COVID-19 in this study, all the mentioned facts need an explanation. In the light of this, the authors’ interpretation of the facts (Lines 261-265) sounds contradictory and unreliable. Indeed, “decrease of dysbiosis during the ICU stay due to administration of dexamethasone” does not agree with a sharp decrease of Chao1 index (Fig. 3) demonstrating the deep dysbiotic disorders. Decrease of Chao1 index in the bronchoalveolar samples (Fig. 3) seems even more dramatic than in the gut ones. So, in this situation a weak reliability of the dysbiosis score used should be noted here. There are some alternatives might be proposed to the authors.  First, the authors’ formula of dysbiotic disorders could be improved based on the available literature or their own development.  Second, the dysbiotic disorders in the patients observed may be described as a complex of qualitative and quantitative changes in microbiomes, including diversity indices, bacterial load, and differential abundance of bacterial taxa. Really it would be more compliant with the current conception of dysbiosis. In addition, I recommend to try Simpson’s index as another score reflecting community evenness. It usually confirms well the drastic qualitative changes accompanied with predominance of one or few bacterial taxa in a microbiome.

Response:

We acknowledge the comments of the reviewer regarding the difference observed between the Chao1 index and the dysbiosis score. Indeed, dysbiosis could be defined at different levels, qualitatively (at the beta and alpha-diversity levels) and quantitatively (microbial load). ICU patients have their microbiome affected at both levels, in particular at ICU admission. The Chao1 index reports the richness of the microbial community and the dysbiosis score reports the compositional distance between patients and healthy controls. We, therefore, hypothesise that antibiotics prophylactic treatment decreases the richness (Chao1) during the ICU stay and that the different treatments the patient received, may affect the composition of the most dominant taxa of the microbial community, as evidenced by the decrease of the dysbiosis score, which relies mainly on the variation of the composition of the most dominant taxa between patients and healthy controls. This hypothesis is supported by the decrease in the dysbiosis score even though the richness and evenness continue to drastically decrease during the ICU stay. Also, we believe that the dysbiosis scores that were found negatively correlated with PaO2/FiO2, at ICU admission, could be used as severity disease biomarkers. In our future studies, as wisely recommended by the reviewer, we will develop a more sophisticated dysbiosis score that will take into account qualitative (alpha and beta diversity) and, if possible quantitative indices.

As the reviewer suggested, we now included the figure with the Simpson index (Supplementary Figure S3). As shown in the figure the Simpson index reflected well the results obtained with the Chao1 index.

3. Discussion covers main findings and results, but does not contain any speculations about role of those bacterial taxa, which have been demonstrated to feature a differential abundance at the statistically significant level except for Streptococcus and Pseudomonas (Line 355). Even a mention of Clostridiales and their beneficial role (Lines 342-345) seems unrelated to the data obtained. Correlations with microbiota in patients with severe COVID-19 are also absent. However, such fragments can improve understanding of the changes in microbiomes recorded in this study and attract more specialists in COVID-19 microecology.

Response:

As suggested by the reviewer, we have now added more speculations related to bacterial species that have been found correlated with ICU severity, based on the PaO2/FiO2 ratio.

“…One of the two bacterial species found negatively correlated with the PaO2/FiO2 ratio belongs to the Actinomycetaceae family, which indicated that it was associated with severe COVID-19 cases. This result is in line with a previous study that showed a positive correlation between Actinomyces viscosus (belonging to the Actinomycetaceae family) with COVID-19 severity (Zuo PMID: 32442562). Actinomyces spp. is known to cause pulmonary actinomycosis, a bacterial lung infection (Farrokh 2014, PMID: 25191495), therefore this species could emerge as a potential trigger or perpetuator of disease severity in the ICU patients”

The reason why we would not add more speculations is related to the long list of bacterial taxa (see supplementary Table S3) involved for instance in the differences between HCs and ICU patients at baseline, which indicates complex interactions between different bacterial groups. On one hand, we observed that among those enriched in ICU patients, most of them were pathogenic types such as Campylobacter, Porphyromonas, Fusobacterium, and Enterobacteriaceae. On the other hand butyrate producers belonging to Clostridiales (Clostridiaceae, Lachnospiraceae, and Ruminococcaceae), known for maintaining intestinal homeostasis, were among the depleted bacterial taxa in ICU patients, as we have mentioned in the discussion section of the MS.

 Minor corrections:

1. Line 108-110. It is unclear, why the antibiotics were prescribed “until the absence of urinary pneumococcal or Legionella antigen detection and a negative bacterial culture”, as though the patients were suffered from legionnaires’ disease or pneumococcal infection or septicemia? Moreover, in Line 106 “a prophylactic dose” of antibiotics is mentioned. This unclarity should be resolved.

Response:

We have now clarified this statement as follows:

"During the ICU stay, 4 (12.5%) ICU-COV(-) and 6 (11.3%) ICU-COV-(+) patients died. As a hospital protocol, to eliminate or prevent possible bacterial co-infection with SARS-CoV2, all critically ill patients received, at admission day,  a prophylactic dose of an antibiotic such as beta-lactam alone (21/32, 66%), and ICU-COV(+) patients received a combination of beta-lactam and macrolides (37/53, 70%), from the ICU admission day until the absence of urinary pneumococcal or Legionella antigen detection and a negative bacterial culture"

2. Line 118. Abbreviations BMI should be deciphered at first mention.

Done

3. Line 147 and below. If the PCR-based bacterial load is interpreted, it never is applied for description of microbiome composition, because its quantitative sense, whereas microbiome composition is a qualitative term. This unclarity should be resolved.

Response:

We agree with the reviewer’s comment and have separated bacterial load from microbiome composition description.  

“ICU-COV(+) patients did not show a significant difference in alpha-diversity but an enrichment of 34 bacterial species, and a depletion of 61 bacterial species compared to HCs (MaAsLin2 method, FDR<0.05, online supplementary Table S4). All ICU patients presented a lower bacterial load compared to HCs.“

4. Line 153 and below. Term “bacterial species” as well as other taxa may be used only if the statistical analyses are performed at different taxonomic levels, including species etc. At the same time, all the supplementary tables look as though statistical analysis was performed at ASV level only, without their clustering into species, genera and other taxa. If there were several analyses after ASVs combining into taxa, at every taxonomic level, it should be indicated clearly in the M&M section, as well as in all captions for figures and tables, where it is necessary.

Response:

Yes, the taxonomic profiling and statistical analysis were done at the species level. We have now included this information in the M&M section.

“…For bacterial profiling, taxonomy was assigned to each ASV using the 16S Greengenes database, the gg_13_8_99 release, which contains 202,421 bacterial and archaeal sequences. Downstream analysis was performed at the species level.”

5. Figure 3. Percentage numbers at all axes in all PCoA plots should be seen clearly. Now, they are absent at axes 3 in the top left and lower right plots.

Response:

Missing percentages numbers have been added to figure 3.

6. Line 206. Unknown Enterobacteriaceae seems curious, taking into account that all main members of this family have been described by the current moment. Possibly, it may be determined by a poor discrimination of the 16S Greengenes database, which last release was in 1999. For such poorly identified ASVs other databases are recommended, such as RDP or Genbank (NCBI), where simple BLAST can substantially specify taxonomy of an OTE or ASV.

Response:

Actually, the updated version of Greengenes we used in the study is from 2013 and is the last version of the database. We believe that the low rate of annotation at the species level is due to the use of a short fragment of the 16S gene (the V4 region). We are currently switching all our other projects to shotgun metagenomic analysis, which will bypass the limitations of poor annotation encountered with the 16S gene.

7. Line 243. “Detection of multi-drug resistant bacteria”, which are not mentioned elsewhere in the manuscript. This unclarity should be resolved.

Response:

Thank you for this observation. We have included this statement by mistake and have now removed it.

8. Lines 398-400 and 408-410 contain absolutely identical statement “All participants…”. This unclarity should be resolved.

Response:

Thank you for pointing out this mistake. We have now removed one of the sentences.

9. Line 419. “All PCR with possible contamination was…” should be exchanged by “All amplicons with possible contamination were…”

Done